# MicroRNA Profiles in Monocyte-Derived Macrophages Generated by Interleukin-27 and Human Serum: Identification of a Novel HIV-Inhibiting and Autophagy-Inducing MicroRNA

**DOI:** 10.3390/ijms22031290

**Published:** 2021-01-28

**Authors:** Tomozumi Imamichi, Suranjana Goswami, Xiaojun Hu, Sylvain Laverdure, Jun Yang, Ju Qiu, Qian Chen, Brad T. Sherman, Weizhong Chang

**Affiliations:** Laboratory of Human Retrovirology and Immunoinformatics, Applied and Development Research Directorate, Frederick National Laboratory for Cancer Research, Frederick, MD 21702, USA; suranjana.goswami@nih.gov (S.G.); xiaojun.hu@usda.gov (X.H.); sylvain.laverdure@nih.gov (S.L.); jyang@mail.nih.gov (J.Y.); ju.qiu@nih.gov (J.Q.); chenq3@mail.nih.gov (Q.C.); bsherman@mail.nih.gov (B.T.S.); weizhong.chang@nih.gov (W.C.)

**Keywords:** macrophages, anti-HIV, microRNA, AB-serum, M-CSF, interferon, IL-27, autophagy

## Abstract

Interleukin-27 (IL-27) is a pleiotropic cytokine that influences the innate and adaptive immune systems. It inhibits viral infection and regulates the expression of microRNAs (miRNAs). We recently reported that macrophages differentiated from human primary monocytes in the presence of IL-27 and human AB serum resisted human immunodeficiency virus (HIV) infection and showed significant autophagy induction. In the current study, the miRNA profiles in these cells were investigated, especially focusing on the identification of novel miRNAs regulated by IL-27-treatment. The miRNA sequencing analysis detected 38 novel miRNAs. Real-time reverse transcription polymerase chain reaction (RT-PCR) analysis confirmed that IL-27 differentially regulated the expression of 16 of the 38 miRNAs. Overexpression of the synthesized miRNA mimics by transfection revealed that miRAB40 had potent HIV-inhibiting and autophagy-inducing properties. B18R, an interferon (IFN)-neutralization protein, partially suppressed both activities, indicating that the two functions were induced via IFN-dependent and -independent pathways. Although the target mRNA(s) of miRAB40 involving in the induction of both functions was unable to identify in this study, the discovery of miRAB40, a potential HIV-inhibiting and autophagy inducing miRNA, may provide novel insights into the miRNA (small none-coding RNA)-mediated regulation of HIV inhibition and autophagy induction as an innate immune response.

## 1. Introduction

MicroRNAs (miRNAs) are endogenous non-coding small RNAs (19–22 nt) that regulate mRNA turnover and translation. The latest miRNA database, miRbase (v22), contains 2656 mature human miRNAs [1]. In general, miRNAs downregulate the expression of target mRNAs. The canonical mechanism of mRNA regulation is based on the binding of miRNAs to the 3′- untranslated region (UTR) of target mRNAs, which represses protein production via mRNA degradation and translational silencing [2]. However, miRNAs interact with the 5′-UTR, coding sequences, or gene promoters [3] and can induce transcription under certain conditions [4]. It has been demonstrated that some miRNAs interact with long non-coding RNAs (lncRNAs) [5,6] and can degrade them directly, thus interfering with lncRNA functionality. Further studies have suggested that the aberrant expression of miRNAs is associated with many human diseases; owing to their secretion into extracellular fluids, miRNAs are considered as potential biomarkers [7,8,9,10,11,12]. Some miRNAs are incorporated into exosomes, where they serve as signaling inducers to regulate cell–cell communication and bystander cell function [10,12,13,14] and some of them are reported to possess the ability to bind to viral RNAs and exert direct antiviral properties [15,16]. Thus, miRNAs can function as multiple-function RNAs that regulate host and viral life cycles.

Interleukin (IL)-27, a member of the IL-12 family of cytokines, plays a multifaceted role in the immune system, in both acquired and innate immune responses [17]. The conventional view of IL-27 function is that IL-27 predominantly acts as an anti-inflammatory cytokine that regulates Th17 and regulatory T cells [18,19]. However, it is known that IL-27 plays a role in not only an anti-, but also a pro-inflammatory cytokine [20,21], and enhances the potential production of reactive oxygen species in monocyte-derived macrophages (MDMs) [22]. In general, in vitro, MDMs are generated from isolated monocytes by culturing the cells with either macrophage-colony stimulatory factor (M-CSF), granulocyte-macrophage-colony stimulatory factor or fetal bovine serum (FBS). Previously, to assess the anti- human immunodeficiency virus (HIV) effect of IL-27, MDMs were infected with HIV-1 and co-cultured in the presence or absence of IL-27. IL-27 treated MDMs were shown to be highly potent inhibitor of HIV-1 [23,24]. In a study aimed to characterize a role of IL-27 during macrophage differentiation, MDMs were generated with a combination of M-CSF and IL-27; MDMs induced with M-CSF alone and a combination of M-CSF with IL-27 were termed “M-Mac” and “I-Mac”, respectively [25]. It has been shown that I-Mac resists infection of HIV, herpes simplex virus (HSV), and influenza virus [25]. Thus, IL-27 is considered a potent agent for treating infected cells and infectious diseases [26]. To elucidate the mechanism underlying the resistance to multi-viral infection in I-Mac, miRNA expression profiles in the cells were investigated and a total of seven novel miRNAs (hsa-miR-6852, -7702, -7003, -7004, -7005, -7006, and miR-SX5) that could potentially target multiple virus genes, including that of HSV-1, were found [27]. Notably, Shabani et al. recently confirmed that hsa-miR-7704 directly targets HSV-1 mRNA and suppresses HSV-1 infection [16].

To elucidate the biological function of IL-27 during monocytes differentiation under physiologically relevant conditions, two types of MDMs using pooled human AB serum were generated: MDMs induced with AB serum alone were designated “AB-Mac” and MDMs induced with a combination of AB serum and IL-27 were named “ABI-Mac”, and then it was found that ABI-Mac are resistant to HIV infection and possess a significantly enhanced autophagy function [28]. This autophagy induction was observed in only ABI-Mac but not in I-Mac. However, the molecular mechanism underlying the autophagy induction and HIV resistance in ABI-Mac remains unclear, and how ABI-Mac resists HIV and induces autophagy is under investigation. In the current study, to further characterize IL-27 function in MDMs, the miRNA profiles regulated by IL-27, focusing on identification and characterization of novel miRNAs in ABI-Mac was investigated.

## 2. Results

### 2.1. Profiles of miRNAs in the ABI-Macs

Cluster of differentiation (CD)14+ monocytes isolated from three independent HIV-uninfected healthy donors (donor identification number (ID #): 89, 90, and 91) were cultured for 7 days in the presence of pooled human AB serum with or without IL-27 and differentiated into two subsets of MDMs. The resulting subsets were designated AB-Mac (AB-89, AB-90, AB-91) and ABI-Macs (ABI-89, ABI-90, ABI-91), respectively. Total cellular RNA was extracted from each cell type, and miRNA sequencing (miRSeq) was conducted (Figure 1A). The obtained miRSeq-reads were cleaned using quality trimming and length filtering. Among the cleaned reads, more than 95% of the bases for all samples indicated a Phred quality score greater than Q30 and yielded between 36 and 62 million reads. After mapping the reads to human reference genome hg38, it was found a total of 1139–1235 and 1119–1252 known miRNAs in AB- and ABI-Macs, respectively (Figure 1B,C). Although significant diversity was expected in the gene lists of the detected known miRNAs among the three donors, because of the presence of donor heterogeneities in the samples, a high-level of similarity in the detected miRNAs was observed: 986 (80–87% of total miRNAs) and 935 (81–83% of total miRNAs) miRNAs were commonly expressed in the setting of AB-Mac (Figure 1B) and ABI-Mac (Figure 1C), respectively.

Next, the similarity in the commonly detected miRNAs between AB-Mac and ABI-Mac from the three donors was compared, to define how IL-27 treatment regulated the miRNA in the donors. A Venn diagram demonstrated that 112 and 61 known miRNAs were uniquely expressed in AB-Mac and ABI-Mac, respectively (Figure 1D; all gene names are listed in Appendix A). Surprisingly, nearly 90% of the detected genes in the list were overlapped, indicating that the expressing known miRNA profile in the setting of AB-Mac was similar to that of ABI-Mac.

### 2.2. Identification of Novel miRNAs in the AB- and ABI-Mac Subsets

Next, using the miRDeep program, identification of novel miRNAs in AB- and ABI-Mac was attempted, and 41 miRNAs were identified: the potent novel miRNAs were termed miRAB1~miRAB41 (Table 1). However, miRAB15, miRAB16, and miRAB38 were identical to our previously reported miRNAs (GenBank accession #: KY994057, KY994062, and MF281431) [29,30], which have not yet been included in the latest version of the database, miRBase 22.1 [1]. The basic local alignment search tool (BLAST) analysis demonstrated that none of the 38 miRNAs were identical with the registered miRNAs in GenBank (as of 31 December 2020). Therefore, in this study, 38 novel miRNAs were identified using the setting of the three donors’ cells. The novel miRNAs were traced back to their possible genomic locations to see whether secondary stem-loop structures could be formed. As shown in Appendix A, these 38 miRNAs were able to form the predicted secondary stem-loop structures.

### 2.3. Confirmation of the Expression of Novel miRNAs Using qRT-PCR

To confirm whether each novel miRNA is expressed in AB-Mac or ABI-Mac, a real-time quantitative reverse transcription polymerase chain reaction (qRT-PCR) was performed using custom-made TaqMan probes for all 38 miRNAs. Since the RNA used in the miRSeq were used for carrying out the microarray assay, to determine target genes (please see Section 2.4 below), in the confirmation assay, to detect each miRNA, a new set of RNA samples were extracted from five independent donors’ cells (Donor 92~96) (Figure 2A). To compare baseline level of each miRAB expression, the cycle threshold (Ct) value of each of miRNA was calculated by comparing with the Ct value of RNU44 in each sample. To avoid misleading the data, the number was subtracted from 40 (normalized gene expression: 40-dCt). As shown in Figure 2B, all novel miRNAs were endogenously detected and some miRABs were differentially expressed in ABI-Mac in the setting of the five donors.

The expression of miRAB4 (10.4 ± 1.3-fold, *n* = 5; *p* < 0.001), miRAB9 (5.3 ± 1.2-fold, *n* = 5; *p* < 0.05), miRAB10 (3.0 ± 0.4-fold, *n* = 5; *p* < 0.05), miRAB12 (16.7 ± 1.87-fold, *n* = 5; *p* < 0.001), miRAB14 (5.9 ± 0.8-fold, *n* = 5; *p* < 0.001), miRAB18 (3.8 ± 1.2-fold, *n* = 5; *p* < 0.01), miRAB19 (4.1 ± 1.4-fold, *n* = 5; *p* < 0.05), and miRAB20 (3.1 ± 0.4-fold, *n* = 5; *p* < 0.05), was significantly upregulated more than 3-fold in ABI-Macs compared to that in AB-Macs, while that of miRAB25 (0.6 ± 0.1-fold, *n* = 5, *p* < 0.01), miRAB27 (0.58 ± 0.1-fold, *n* = 5, *p* < 0.05), miRAB29 (0.3 ± 0.02-fold, *n* = 5, *p* < 0.001), and miRAB30 (0.3 ± 0.1-fold, *n* = 5, *p* < 0.01) was significantly downregulated.

### 2.4. Prediction of Potential miRNA Target Genes

Since the expression of miRAB4, miRAB9, miRAB12, miRAB14, miRAB29, and miRAB30 in ABI-Mac of Donor 92~96 was significantly changed by more than 4-fold compared to that in the AB-Mac by qRT-PCR (*p* < 0.001) (Figure 2B), the identification of potential target genes for each miRNA was attempted using four target prediction tools (TargetScan [31], MR-microT [32], miRanda [33], and miRDB [34]), as described in the Materials and Methods. Commonly predicted genes by more than three of the tools were identified (Figure 3). All genes are listed in Appendix A.

To assess whether the predicted genes was downregulated by the expression of each novel miRNAs, microarray assay was conducted using the aliquots RNA of AB and ABI-Mac (Donors 89~91) that was used for miRSeq in Figure 1. Then the expression levels of each predicted gene in the microarray data were analyzed. As summarized in Table 2, six predicted target genes were found to significantly decrease (*p* < 0.05) by more than 2-fold in ABI-Mac. miRAB4 and miRAB29 targeted Dipeptidyl peptidase 4 (DPP4), while three miRABs (miRAB9, miRAB14, and miRAB29) targeted RAB6B (member RAS oncogene family 6B). Interestingly, C-X-C motif chemokine ligand 9 (CXCL9), targeted by miRAB14, was found to increase by 15-fold in ABI-Mac. Later, we conducted a real-time RT-PCR assay to confirm the increased in the gene expression using five new independent samples of AB- and ABI-Mac (Donor 97~101); the expression of CXCL9 in ABI-Mac was changed by 0.05~250-fold compared to that in AB-Mac (Appendix A), indicating that miRAB14 expression in ABI-Mac may have little effect on the CXCL9 expression, and donor-to-donor heterogeneity may affect the expression level of CXCL9 in ABI-Mac.

Prediction analysis of viral targets by all novel miRNAs was also conducted as previously described [27,29]. All novel miRABs, except miRAB13, potentially targeted multiple classes of virus RNAs (potential targeted viruses were predicted using miRanda; the virus names are listed in Appendix A); miRAB13 does not target any viruses. Of note, none of miRAB, including miAB13, target the coronavirus disease 2019 (COVID-19) virus.

### 2.5. Evaluation of the Biological Functions of the Novel miRNAs

In the next step, the anti-HIV activity of the 38 miRABs was assessed. We have been investigating the mechanism of HIV replication and antiviral effects of cytokines and microRNA profiles using M-Mac [22,23,25,27,30]. To obtain a comparable data set using the 38 novel miRAB, a series of assay was performed using M-Mac. A total of 38 individual miRAB mimics were chemically synthesized and transfected into M-Mac (Figure 4A). The transfected cells were infected with either a replication-competent virus (HIVAD8) or a replication-incompetent HIV pseudotyped virus encoding luciferase (Luc) gene (HIVLuc-V). As shown in Figure 4B, miRAB3 and miRAB40 suppressed HIVLuc-V infection. MiRAB3 significantly inhibited only HIVLuc-V infection but not HIVAD8 replication; it suppressed HIVLuc-V infection by 87 ± 6.8% (*p* < 0.05), compared to miRCtrl-transfected cells. In contrast, miRAB40 transfection significantly repressed infection of both viruses; it inhibited HIVAD8 replication and HIVLuc-V infection by 91 ± 7.0% (*p* < 0.05) and 94 ± 12.8% (*p* < 0.05), respectively, suggesting that the inhibitory mechanism of miRAB3 differs from that of miRAB40. A number of novel miRABs modestly enhanced or inhibited HIV replication/infection; however, none of them showed significant changes in the activity.

We previously reported that ABI-Macs resist HIV infection and induce a high level of autophagy [28], thus we hypothesized that some of miRABs discovered in ABI-Mac may be potent autophagy inducers. To address the hypothesis and further characterize the function of each miRAB, an autophagy assay was performed. To define a correlation with anti-HIV effects, the autophagy assay was also carried out using M-Mac. Autophagy function was determined by staining of autophagosomes in each of the miRAB-transfected MDMs with Cyto-ID dye (Enzo Life Sciences) and quantified by image analysis using the Fiji ImageJ software [35] (Figure 4C), As shown in Figure 4D, autophagy was significantly induced in miRAB3- and miRAB40-transfected cells, but not in other miRAB-transfected cells. Compared to miRCtrl-transfected cells, miRAB3 and miRAB40 transfection enhanced autophagy by 3.3 ± 0.8 (*n* = 4; *p* < 0.05) and 31.2 ± 15.6 (*n* = 4; *p* < 0.05)-fold, respectively (Figure 4E demonstrates representative images of the autophagy induction by miRAB3 and miRAB40 transfection in M-Mac compared to miRCtrl-transfected M-Mac). Thus, miRAB3 and miRAB40 were found to potentially possess both HIV-inhibitory and autophagy-inducing activities. Transfection of miRAB40 mimic induced autophagy more than miRAB3 and inhibited both HIVAD8 and HIVLuc-V infection by more than 90%; we focused on miRAB40 in subsequent studies.

### 2.6. Investigation of the Mechanism Underlying miRAB40-Mediated Anti-HIV and Pro-Autophagy Properties

It is reported that transfection of small RNAs (e.g., miRNA and small interfering RNA) activates the innate immune system and induces type-I and -III interferons (IFNs) [36,37,38,39]. To identify whether miRAB40 induces IFNs, M-Mac (donor 111~114) were transfected with miRCtrl or miRAB40, and then type-I and -III IFNs in culture supernatants were quantified using enzyme-linked immunosorbent assay (ELISA) kits (Figure 5A). The transfection of miRAB40 mimic significantly induced type-I IFNs compared to untreated cells: IFN-αs and IFN-β were induced by 26.5 ± 7.2 pg/mL (*n* = 4; *p* < 0.01) and 6.7 ± 2.2 pg/mL (*n* = 4; *p* < 0.01), respectively (Figure 5B). Transfection of miRCtrl also elicited a low level of IFN-αs (0.9 ± 0.4 pg/mL) and IFN-β (0.7 ± 0.4 pg/mL). Of note, IFN-λ induction could not be detected from miRCtrl- and miRAB40-transfected cells.

Type-I IFNs (IFN-α and β) inhibit HIV, but their role in autophagy induction is controversial [40]. To address the correlation among the IFN production, HIV inhibition, and autophagy induction, a recombinant B18R, which is a soluble vaccinia virus-encoded type I-IFN receptor (type-I IFN neutralizing protein) was employed [41]. As a positive control for neutralization, 100 units/mL of recombinant IFN-α (protein concentration: 207 pg/mL, specific activity: 4.83 × 10^8^ units /mg; PBL Assay Science) was used (Figure 5C). In the absence of B18R, IFN-α and miRAB40 inhibited HIV infection by 89.2 ± 7.3% (*n* = 3) and 99.7 ± 0.2% (*n* = 3), respectively, compared to untreated cells (Figure 5D). In the presence of 1 µg/mL of B18R, the IFN-α-mediated inhibition was abolished, and HIV infection was restored to a comparable level (105.8 ± 4.4%; *n* = 3) as untreated cells. In contrast, miRAB40-mediated HIV infection was partially restored (34.4 ± 21.1%; *n* = 4, *p* < 0.05), indicating that miRAB40-induced anti-HIV action is mediated by IFN-α-dependent and -independent mechanisms. In the autophagy assay (Figure 5E), 100 units/mL (207 pg/mL) of IFN-α induced autophagy by 11.5 ± 4.8-fold (*n* = 4), compared to untreated control cells (Figure 5F) and this induction was neutralized by B18R. On the contrary, miRAB40 transfection significantly induced autophagy by 75.8 ± 30.5-fold (*n* = 4) (*p* < 0.01), even though IFN-α concentration in the culture supernatants of miRAB40-transfected cells was lower than that in recombinant IFN-treated cells (Figure 5B), and B18R partially suppressed this induction by 71.0 ± 18.6%. This result indicated that miRAB40 induces autophagy in an IFN-dependent manner; however, the induction may involve an uncharacterized factor that synergizes with IFN to enhance autophagy. It was speculated that miRAB40-transfected cells may secrete a soluble autophagy-inducing factor in the culture medium that may be involved in autophagy induction. To elucidate this hypothesis, cell-free culture supernatants from miRAB40-transfected cells were collected and supplemented to the culture of untreated M-Mac, which were then assayed for autophagy. The supernatants were found to have no impact on the autophagy induction (Appendix A), suggesting that the unknown factor(s) is a cytosolic factor rather than soluble factors.

To identify the uncharacterized cytosolic factor(s) possessing anti-HIV and autophagy-inducing activities in miRAB40-transfected cells, target prediction analysis was performed using the four prediction tools as described above. A total of 11 genes and 177 genes were commonly predicted as potential targets of miRAB40 by four and three of the prediction tools, respectively (Appendix A, all genes are listed in Appendix A). To determine which genes were potentially involved in anti-HIV activity or autophagy induction in M-Mac, the 188 predicted genes were compared with the known host-dependency factors (HDFs) for HIV infection [42,43,44] or the three autophagy databases: Human Autophagy Moderator database (HAMdb; http://hamdb.scbdd.com), Human Autophagy Database (LuHADb; http://autophagy.lu), and Autophagy Database (Autophagy; http://www.tanpaku.org/autophagy). Among a total of 932 HDFs, only six genes—ETS proto-oncogene 2 transcription factor (ETS2), DEAH-box helicase 15 (DHX15), 3-hydroxyisobutyryl-CoA hydrolase (HIBCH), transducin beta-like 1 X-linked (TBL1X), solute carrier family 24 member 1 (SLC24A1), and mitogen-activated protein kinase kinase kinase 7 (MAP3K7)—were identified as potential targets of HDFs (Appendix A). Similarly, among a total of 1852 genes reported as autophagy regulatory factors (ARFs) in the autophagy databases, six genes—phosphatidylinositol 3-kinase catalytic subunit type 3 (PIK3C), RAB9A, member RAS oncogene family (RAB9A), n-RAS proto-oncogene (NRAS), salt inducible kinase 1 (SIK1), integrin subunit beta 1 (ITGB1), and cell proliferation regulating inhibitor of protein phosphatase 2A (CIP2A/KIAA1524)—were identified as potential target of ARFs (Appendix A).

To clarify changes in the expression of the predicted genes in miRAB40-transfected M-Mac (Donor 119~121), a real-time RT-PCR assay was conducted using gene-specific probes. None of the target genes of HDFs and ARFs were significantly downregulated in the setting of the miRAB40-transfected cells (Table 3). Only SLC24A1 (2.5 ± 0.5-fold, *n* = 3; *p* < 0.05) was significantly upregulated. To confirm the increase in the expression of SLC24A1 protein corresponding to the gene expression, western blotting analysis was conducted using cells from three donors (Donor 111, 121, 122). However, protein levels in miRAB40-transfected M-Mac were not significantly changed (Appendix A). Taken together, the targets of miRAB40 associated with anti-HIV and pro-autophagy activity in M-Mac were not identified.

## 3. Discussion

IL-27 is an antiviral cytokine that inhibits infection of HIV, HSV, and other viruses [23,25,45,46]. We previously reported that IL-27-treated MDMs and CD4+ T cells exhibit HIV resistance [25,47] and identified a total of 44 novel microRNAs in the IL-27-treated cells [27,29,30]. Recently, we reported that IL-27 can induce a new subtype of MDMs (ABI-Mac), which possess a high level of autophagy function, compared to AB-Mac [28]. In the current study, we compared the miRNA expression profiles between AB-Mac and ABI-Mac from normal healthy donors and discovered 38 novel miRNAs that were endogenously detected in the setting of both cell types in the donor cells tested (Figure 2B). On contrary, the novel miRNAs had not been previously detected in M-Mac, CD4+ T cells, and Dendritic cells, even those cells were treated with IL-27 [27,29,30], suggesting that the expression level of the novel miRNAs may be regulated by 1) cell type specific manner, and/or 2) donor-to-donor dependency. In our study, the novel miRNAs were detected in macrophages from randomly chosen donors; however, to determine the expression level of the miRNAs in cell type specificity, in general, we need to perform a population difference analysis using robust sample size considering gender, age, and genetic factor in donors. A study of the longitudinal analysis of the expression of novel miRNAs in cells from HIV infected patients and a correlation analysis between the expression level and viral loads in HIV-infected donors would provide new insights in a mechanism of the regulation of miRNA expression in virus infection and in a development of novel therapy using small RNA.

Biological functional analysis using miRNA mimics revealed that miRAB40 possessed both HIV-inhibiting and autophagy-inducing properties in tested cells. ELISA results showed that miRAB40 preferentially induced IFN-αs (approximately 30 pg/mL) rather than IFN-β or -λs. A Type-I IFN-neutralization protein, B18R partially suppressed the anti-HIV effect and autophagy induction; thus, it was concluded that both anti-HIV activity and autophagy induction by miRAB40 was caused by IFN-dependent and -independent mechanisms. Since the activities induced by miRAB40 (which induced nearly 30 pg/mL IFN-αs and 10 pg/mL IFN-β in the culture) were robust and higher than that by 100 units/mL of recombinant IFN-α alone; it was equivalent with 207 pg/mL IFN-α, according to the specific activity of the lot of IFN-α that we used in the assay. Therefore, the miRAB40-promoted activities may have been synergistically induced by Type-I IFNs and the unknown factor(s). We attempted to define the underlying mechanism using bioinformatics tools, but we were unable to identify it in this study. There are conflicting reports on the association between IFN production and autophagy induction and whether type-I IFN induces or inhibits autophagy [40,48,49]. Taken together, these reports indicate that the relationship between autophagy and type-I IFN production may depend on a number of factors, including the cell type and the specific receptors involved [40], despite the fact that our data indicated that IFN expression and anti-autophagy induction are correlated in the context of miRNA treatment. To elucidate the specificity of the biological function in miRAB40 mimic, we need further study using other miRNAs, e.g., miRAB3 and different cell types (e.g., M-Mac, AB-Mac, and ABI-Mac).

The miRAB3 mimic also inhibited HIV, it suppressed only HIVLuc-V infection, but not HIVAD8 replication. HIVLuc-V assay is conducted on the second day after infection to detect HIV in the early stage of the viral cycle, while HIVAD8 replication assay is performed in a 14-day-culture to define the impact of miRNA on the late stage of the viral life cycle during multiple rounds of infection. Since miRAB3 only inhibited HIVLuc-V infection, the miRNA interferes in a step that does not affect viral replication. A pilot study demonstrated that miRAB3 transfection (using M-Mac from donor 111, 123~125) also induced IFN-αs (9.5 ± 8.8 pg/mL, *n* = 4), which was a relatively lower amounts compared to miRAB40. It is plausible that the induced IFN-αs had no impact on HIVAD8 replication. Further analysis is needed to identify the mechanism of the anti-HIV effect by miRAB3. As shown in Figure 4B, a number of novel miRABs modestly enhanced or inhibited HIV replication/infection, however, none of them significantly changed the activities due to a low level of reproducibility, suggesting that target genes of those miRNAs may express donor dependent manner. It may be of interest to investigate the mechanism of the modification using cells from the miRAB responders.

It is known that exogenous RNAs (transfected RNA, viral RNA from the infected RNA viruses) induce type-I IFN via cytosolic RNA sensors such as Retinoic acid-inducible gene I (RIG-I) or melanoma differentiation-associated gene 5 (MDA5) [50]. Although miRNA mimics were considered incapable of inducing the innate immune response because they are short and structurally similar to natural miRNAs, some miRNAs are known to regulate IFN induction [30,36,37,51,52]. In the current study, we reported that miRAB40 is a novel potent IFN-α-inducing miRNA that is 24 nt-long. However, other miRNAs (miRAB15, miRAB22, and miRAB33) that were also 24 nt-long (Table 1) had no significant IFN induction. Therefore, the miRAB40-mediated IFN induction may occur in a sequence-dependent manner. It is reported that some miRNAs interact with lncRNAs [5,6]. Thus, it is possible that miRAB40 may affect function(s) of a lncRNA and regulate IFN induction. It would be interesting to define the mechanism of the IFN induction by miRAB40. MiRAB40 locates on chromosome 1 between 221076659 and 221076720 nt as an intergenic miRNA. The expression and regulation of transcription of the host gene near the location remains unclear. As shown in Figure 2, the expression of endogenous miRAB40 in AB-Macs were relatively lower than other miRNAs and was not regulated by IL-27-treatment. Defining the mechanism of induction of the miRNA may also find a regulatory mechanism of autophagy induction by the miRNA. Future studies should be performed to define the mechanism of the IFN induction via miRNAs.

When ABI-Mac was induced with human AB serum in the presence of IL-27 for 7 days, it was expected that IL-27 treatment would polarize the macrophages into a unique subset, even though it was thought that there might be some donor differences in the cellular responses to IL-27 and that the miRNA profiles of the ABI-Macs would be significantly different from those of the AB-Mac. However, as shown in Figure 1D, nearly 90% of detected known miRNAs were comparable between the settings of two cell types. Since those results were obtained from the three donors, it is limited in the interpretation. To address the function of IL-27 in general, further studies are needed. A study of the expression profile and a longitudinal analysis of the novel miRNAs in HIV infected patients’ cells and a correlation analysis between the expression profile and viral loads in individual patient would also provide new insight in developing a novel therapy using small RNAs.

In conclusion, 38 novel miRNAs were detected in the AB serum and IL-27-induced macrophages. At this moment, regulatory mechanism of the expression of each miRNA in each individual is not clear; however, one of novel miRNAs–miRAB40–was identified as a potential HIV-inhibiting and autophagy-inducing miRNA. In general, it is known that the expression of miRNAs modulate many cellular functions [53,54,55]. Because IL-27 functions in many different cell types, not only immune cells [18], but also hepatocytes [46] and tumor cells [56,57], it is considered to be an immunotherapeutic agent [26]. Therefore, it is important to understand IL-27-mediated changes in miRNA profiles in these cell types; further studies may provide new insights into IL-27 immunotherapy for infectious diseases and cancer.

## 4. Materials and Methods

### 4.1. Generation of Cells and Viruses

Peripheral blood mononuclear cells (PBMCs) were isolated from healthy HIV-uninfected donor’s apheresis as previously described [23]. CD14+ monocytes were isolated from the PBMCs using MACS CD14 MicroBeads (Miltenyi Biotec, Auburn, CA, USA). The monocytes were differentiated into macrophages in D10 medium (D-MEM medium (Thermo Fisher Scientific, Waltham, MA, USA) supplemented with 10% FBS (HyClone, GE-Health Care, Chicago, IL, USA), 10 mM 4-(2-hydroxyethyl)-1-piperazineethanesulfonic acid (HEPES) (Quality Biology, Gaithersburg, MD, USA) and 5 µg/mL of Gentamycin (Thermo Fisher) with 10% pooled human AB serum (Gemini Bio, West Sacramento, CA, USA) [23] in in the presence or absence of 100 ng/mL of IL-27 (R&D systems, Minneapolis, MN, USA) for 7 days. Differentiated macrophages were maintained in D10 medium. M-CSF (R&D Systems)-induced MDMs were prepared as previously described [25]. Replication competent HIV(HIVAD8) and VSV-G-pseudotyped HIV-luciferase virus (HIVLuc-V) were prepared by transfection of pHIVAD8 [58] and co-transfection of co-transfection of plasmids pHIVLuc [59,60], respectively, as previously described [25].

### 4.2. Preparation of RNA and Quality Control (QC) for RNA and RNA Seq

Total RNA was extracted using the Qiagen miRNeasy kit (Qiagen, Hilden, Germany). Prior to RNA sequencing, samples were subjected to analysis by the Agilent Bioanalyzer RNA Nano chip and the small RNAchip (Agilent Technologies, Santa Clara, CA, USA) to confirm RNA purity and quality. Six microRNA libraries were prepared using the NEBNext Multiplex Small RNA Library Prep for Illumina protocol (New England BioLabs, Ipswich, MA, USA) according to the manufacturer’s instructions and sequenced on the Hiseq 2500 using v4 chemistry for single end sequencing (San Diego, CA, USA). RNA sequencing (miRSeq) was performed by the National Cancer Institute-Center for Cancer Research Sequencing Facility (Frederick, MD, USA). All novel small RNA sequences have been deposited into the National Center for Biotechnology Information (NCBI) Sequence Read Archive (SRA) database under accession number: MF281454~MF28149.

### 4.3. Read Processing and Mapping

After quality control of sequencing reads were done, adaptors were trimmed using the Cutadapt program [61]. The remaining reads were mapped to the human reference genome hg38 using the Burrows–Wheeler aligner (BWA) (v0.7.10-r789) [62] with one mismatch as recommended by others [63] for miRNA mapping. Data were compressed with SAMtools [64] to bam format followed by intersecting human genomic miRNA coordinates from miRbase (v21 hsa.gff3) [65] using BEDTools muticum [66].

### 4.4. Identification of Novel miRNAs

Identification of miRNAs was performed using the tool miRDeep [67]. Prior to the analysis, the read count matrix was normalized using EdgeR package 3.16.5 [68,69]. The significant miRNAs were selected based on fold changes >1.5 or <−1.5, false discovery rate (FDR) <0.05 and at least one sample count. A list of the miRNA candidates from each sample were merged by their genomic coordinates using BEDTools intersect [66]. The genomic regions were annotated by GENCODE v24 [70] and ANNOVAR [71] with hg38 databases. The regions having other RNAs, like snoRNA, snRNA, tRNA, rRNA, Y-RNA, etc., were excluded at this step. Furthermore, the precursors were loaded into RFam [72] (v12.1, available online: http://rfam.xfam.org/) for RNAs characterization. The precursor minimum free energy (MFE) was computed by RNAFOLD [73] with a cutoff value of ≤−20 kcal/mol. The mature sequence guanine-cytosine (GC) content was calculated with an in-house Perl script with a cutoff value of ≤80%. The interesting candidates were further identified by their coordinates with tracks supplied by the University of California Santa Cruz (UCSC) genome browser (available online: https://genome.ucsc.edu/) and their RNA secondary structures from miRDeep2 [67].

### 4.5. Prediction of Target mRNA of miRNAs

Potential targets of the miRNAs of interest were predicted using four prediction tools: TargetScan [31], MR-microT [32], miRanda [33], and miRDB [34]. In the prediction analysis, the following cutoffs were used: miRDB sore was >80, MR-microT score cutoff was > = 0.7, miRanda minimum free energy < = −20 kcal/mol and TargetScan cutoff context + + score < −0.3. To demonstrate overlapping genes among the predicted targets, a Venn diagram analysis was performed using a tool, http://bioinformatics.psb.ugent.be/webtools/Venn/generated. Prediction of viruses targeted by miRNAs was conducted as previously described [29]. Moreover, 745 human viral reference genome sequences were downloaded from the National Center for Biotechnology Information (NCBI) viral genome browser (available online: http://www.ncbi.nlm.nih.gov/genomes/GenomesGroup.cgi?taxid = 10239&opt = Virus & sort = genome) and were scanned by miRanda [33] with a minimum free energy <−20 kcal/mol using 38 novel miRNAs. The viral binding sites were further filtered with a pairing score > 100 as potential miRNA targets.

### 4.6. Microarray Data Analysis

Aliquots of RNAs that was used for miRSeq were used for performing Affymetrix GeneChip assays. Microarray assay was conducted as previously described [25]. Briefly, Affymetrix Human GeneArray 2.0 (Affymetrix, Thermo Fisher Scientific) was used and labeling assay was performed by following the manufacturer’s protocol. Prior to analyze microarray raw data, a robust multi-array average (RMA) exam [74] was conducted on all data as a normality test, and then the QC passed data were analyzed using Partek Genomics Suite (St. Louis, MO, USA). A 2-way ANNOVA statistical analysis was performed on genes of interests to determine differentially expressed genes between AB- and ABI-Mac.

### 4.7. Cross Analysis of Predicted miRNA Targets and Differently Expressed Genes

The predicted potential targeted genes and gene lists from the microarray analysis were cross-analyzed using the Venn diagram. The gene expression profile with the *p* value and fold change between AB and ABI-Mac comparison was combined with the predicted target genes by the official gene symbol (some Ensembl IDs target multiple genes) for each of the interested miRABs.

### 4.8. Real-Time PCR

To quantify relative expression of novel miRNA and host mRNA of interest, qRT-PCR was performed using the TaqMan MicroRNA Assays (Thermo Fisher Scientific) and TaqMan Reverse Transcription Reagents (Thermo Fisher Scientific), respectively. All assays were run on the iCycler real-time PCR detection system (Bio-Rad, Hercules, CA, USA). Gene-specific primers and probes for the novel miRNA were custom made by Thermo Fisher Scientific. A total RNA was extracted from cells using QIAzol reagent (Qiagen, Germantown, MD, USA) as previously described [45]. Reverse transcription (RT) of RNA, followed by real- time PCR was performed using 10 ng and one μg of RNA for miRNA and mRNA, respectively [75]. The level of gene expression was normalized to RNU44 for miRNA and glyceraldehyde 3-phosphate dehydrogenase (GAPDH) for mRNA using probes for each gene (Thermo Fisher Scientific) to obtain comparative data sets with our previous works [25,27,30]. To demonstrate relative amounts of endogenous expression of each miRNA, results display using the 40-dCT method. The relative expression of each mRNA was calculated using the 2−ΔΔCt method [76].

### 4.9. Transfection of miRNA Mimic

Each novel miRAB mimic was synthetized by Thermo Fisher and then transfected into M-CSF-induced MDMs (50 × 10^3^ cells/96-well). Briefly, 50 × 10^3^ cells of MDMs were seeded in 96-well plates and cultured overnight at 37 °C. A final 10 nM miRNA mimic was transfected into MDMs using RNAiMAX (Thermo Fisher Scientific) according to a manufacture protocol and then incubated for 48~72 h prior to subsequent assays.

### 4.10. HIV Replication Assay

HIV replication was monitored using HIVAD8 as previously described [23,77]. Briefly, the miRNA mimic-transfected MDMs were infected with HIVAD8 at 5000 tissue culture infectious dose 50 (TCID_50_)/1 × 10^6^ MDMs cells for 2 h and then cultured for 14 days. All assays were performed in quadruplicate and a half of the culture medium was changed every 3 to 4 days. HIV replication was monitored by p24 concentration in culture supernatants using a p24 antigen capture kit (PerkinElmer, Wellesley, MA USA) [23].

### 4.11. HIV Infection Assay

HIV Infection assay was performed using HIVLuc-V as previously described [25,47]. Briefly, the miRNA mimic-transfected MDMs were infected with 100 ng p24/mL for 2 h and then cultured for 48 h. Infection activity was monitored by measuring Luciferase activity with a Bright-Glo Luciferase Assay System (Promega, Madison, WI, USA) [47].

### 4.12. Autophagy Assay

MDMs in 96-well plates (50 × 10^3^ cells/well) were transfected with 10 nM each miRNA mimic as described above. The transfected cells were cultured for 72 h and then autophagosome staining was performed using the Cyto-ID^®^ reagent (Enzo Life Sciences, Farmingdale, NY, USA) and Hoechst 33342 as a counterstain, as per manufacturer instructions. Briefly, culture medium was removed and replaced with 5% FBS-supplemented Phosphate-buffered saline (PBS) containing both stains. Cells were then incubating at 37 °C for 30 min prior to imaging. Autophagosome (2 × 2 composite) images were taken on a Zeiss Axio Observer.Z1 motorized microscope (Carl Zeiss Microscopy, White Plains, NY, USA) at a 10× magnification. Quantification analysis of each image was performed using the Fiji Image J software (National Institutes of Health, Bethesda, MD, USA) [35]. For each channel, a background threshold was set up in order to create 8-bit masks. For the green channel, the total stained area was retrieved as a measure of autophagosome staining, while a particle counts following watershed processing of the blue channel gave the corresponding cell number, then resulting values display a total cell for each experimental condition. The results present autophagy induction in each miRAB-transfected cell compared to that in miRCtrl (C)-transfected cells by means of fold change (FC) ± SE (*n* = 4).

### 4.13. Enzyme-Linked Immunosorbent Assay

The culture supernatants from miRNA-transfected cells were collected after 72 h post-transfection and concentrations of IFN-α, β and λ were measured using the VeriKine-HS Human IFN-α All Subtype ELISA Kit (PBL Assay Science, Piscataway NJ, USA), the VeriKine-HS Human IFN-β ELISA Kit (PBL Assay Science), and the human IL-29 ELISA kit (Invitrogen, Thermo Fisher Scientific), respectively, as per manufacturer’s instructions.

### 4.14. Neutralization of Type-I IFN Using B18R

In the absence or presence of 1 µg/mL B18R (R&D Systems), M-Mac were transfected with 10 nM miRNAs as described above and cultured for a total of 72 h. As a positive control for the B18R effect, IFN-α (100 units/mL) (PBL Assay Science, Piscataway, NJ, USA) was added to the un-transfected MDMs.

### 4.15. Statistical Analysis

Results from biological functional assays were representative of at least three independent experiments. The values are expressed as means ± SE. Statistical significance was determined by unpaired two-tailed Student’s *t*-test using Prism 8 software (GraphPad, San Diego, CA, USA). *p* values < 0.05 were considered to indicate a statistically significant difference between the experimental groups (*: *p* < 0.05, **: *p* < 0.01, ***: *p* < 0.001). Analysis of microarray was performed the 2-way ANOVA statistical analysis.

## Figures and Tables

**Figure 1 ijms-22-01290-f001:**
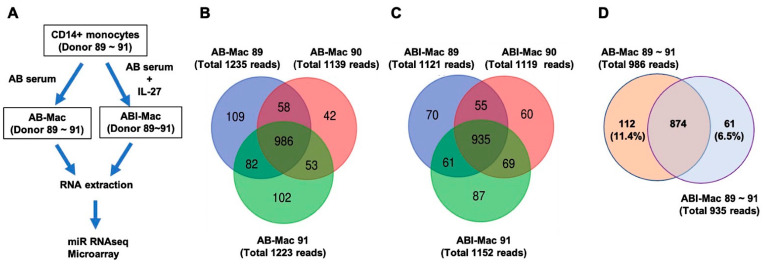
Comparison of known miRNA profiles between AB- and ABI-Macs. (**A**) The diagram depicts the protocol for the preparation of AB- and ABI-Mac from fresh CD14+ monocytes of three independent donors (Donor 89~91) using human AB serum without or with IL-27. (**B**,**C**) Known miRNA profiles in AB-Mac (**B**) and ABI-Mac (**C**) were compared among the donors’ data and the Venn diagram analysis was carried out. (**D**) A profile of 986 known miRNAs commonly expressed in the three donors’ AB-Mac was compared with that of 935 common miRNAs of ABI-Mac and the Venn diagram analysis was performed.

**Figure 2 ijms-22-01290-f002:**
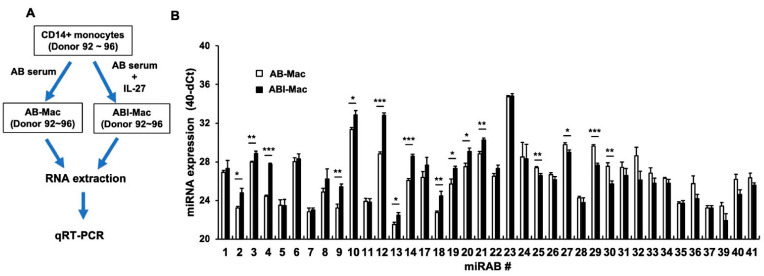
Confirmation and comparison of the expression of novel miRNA. (**A**) A diagram depicts the protocol for the qRT-PCA assay. Total cellular RNAs from AB and ABI-Mac from five independent donors (Donor 92~96) were extracted and then real-time qRT-PCR was performed using specific probes for each novel miRNA. (**B**) RNU44 was used as an internal control, and each miRNA expression was calculated delta-Ct (dCt) using Ct values in each novel miRNA and the RNU44, and then results are described 40-dCT. Results show mean of dCt ± standard errors (SE) (*n* = 5). Gene expression level of each miRAB in ABI-Mac was compared to that in AB-Mac. *, **, and *** indicate *p* values are < 0.05, < 0.01, and < 0.001, respectively.

**Figure 3 ijms-22-01290-f003:**
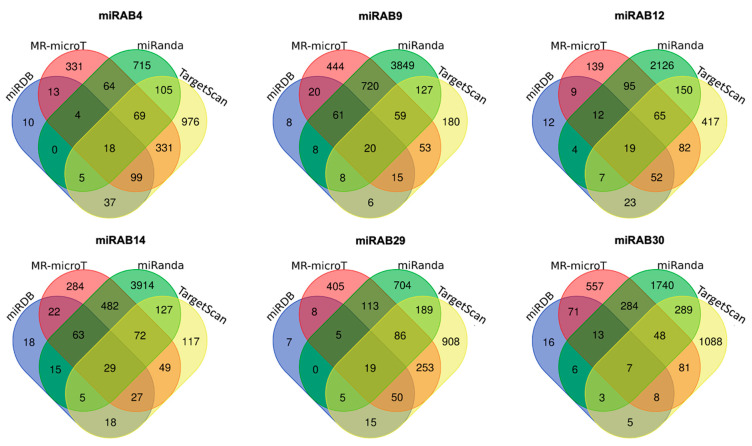
Target prediction analysis. Target mRNAs for each miRAB4, miRAB9, miRAB12, miRAB14, miRAB29, and miRAB30 were performed using miRDB, MR-microT, miRanda, and Target Scan programs. Cut off criteria for each tool were score > 80 for miRDB, score > = 0.7 for MR-microT, max E < = −20 for miRanda, and context ++ < −0.3 for TargetScan. To identify commonly predicted target genes, the Venn diagram analysis was performed.

**Figure 4 ijms-22-01290-f004:**
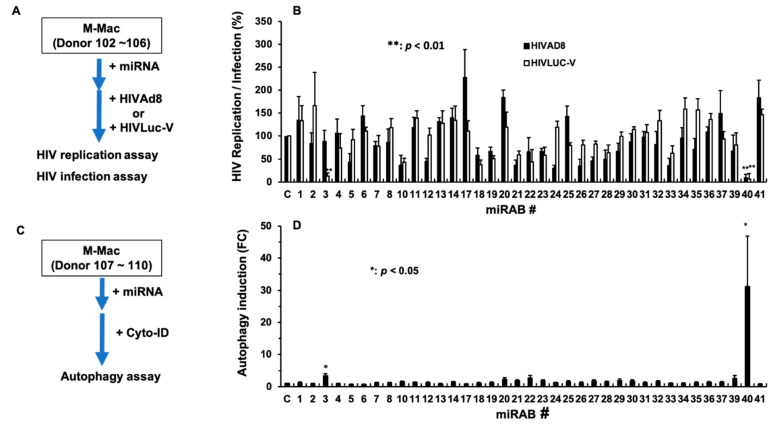
Evaluation of HIV-inhibiting and autophagy-inducing activities. M-Macs were transfected with final 10 nM of chemically synthesized novel miRNA mimics and then cultured for three days at 37 °C. (**A**) Diagrams depict protocol of HIV assay. The miRNA-transfected M-Mac from five independent donors (Donor 102~106) were infected with HIVAD8 or HIVLuc-V virus, and then cultured for 14 days for HIVAD8-infected cells and for 48 h for HIVLuc-V-infected cells as described in the Materials and Methods. (**B**) HIV replication in culture supernatants was monitored with an HIV antigen capture ELISA kit, and HIV infection was determined using cell lysate of the infected cells with the Bright-Glo Luciferase kit. All experiments were performed quadruplicate. Data indicate relative amounts of HIV replication activity in each miRAB-transfected cell compared to that in miRCtrl-transfected cell by means ± SE. (**C**) Diagrams depict protocol of autophagy assay. To detect autophagy induction, autophagosomes in the miRNA-transfected cells from four independent donors (Donor 107~110) were stained as described in the Materials and Methods. (**D**) Each assay was conducted duplicate. Data indicate relative amounts of autophagy activity in each miRAB-transfected cell compared to that in miRCtrl [C]-transfected cell by mean of fold change (FC) ± SE. (**E**) An autophagosome staining image of M-Mac (Donor 107) transfected with either miRCtrl, miRAB3, or miRAB40. Images were taken at 10× magnification using a Zeiss Axio Observer.Z1 motorized microscope (Carl Zeiss Microscopy), as described in the Materials and Methods, and used for quantification of autophagy induction. *, **, and *** indicate *p* values are < 0.05, < 0.01, and < 0.001, respectively. Scale bars show 100 μm.

**Figure 5 ijms-22-01290-f005:**
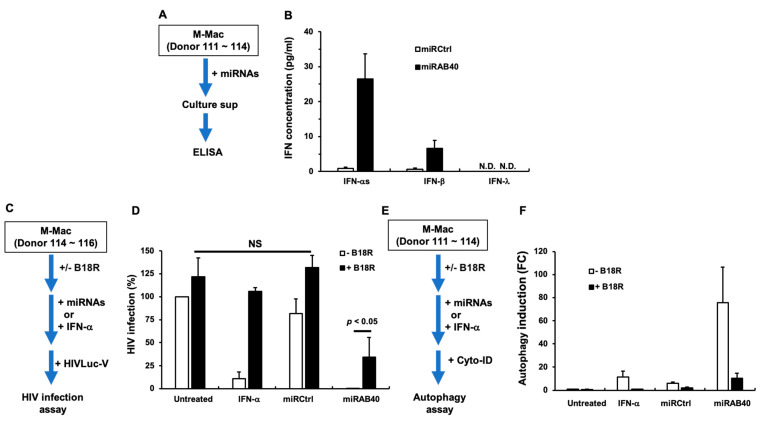
Evaluation of interferon (IFN)-inducing activity and correlation of IFN induction and anti-HIV and autophagy induction. (**A**) M-Macs from four independent donors (Donor 111~114) were transfected with miRCtrl or miRAB40 and cultured for 72 h. The culture supernatants were collected. (**B**) Each IFN concentration in the collected supernatants was measured using subtype-specific ELISA kit. Data represent mean ± SE. Detection limits of IFN-αs, -β and -λs were 1.25, 1.2, and 15.6 pg/mL, respectively. (**C**) To determine a role of IFNs on HIV inhibition, M-Mac from three independent donor (Donor 114~116) were transfected with miRCtrl or miRAB40 in the presence or absence of 1 μg/mL of B18R for three days and then infected with HIVLuc-V as described in the Materials and Methods. (**D**) HIV infection was quantified using with the Bright-Glo Luciferase kit. As a positive control for the neutralization of B18R effect, untransfected cells were treated with 100 units/mL of IFN-α2b (R&D systems) for 3 days and then infected with the virus. Data indicate % of HIV infection compared with untreated cells from four independent assays using means ± SE. (**E**) To determine a role of IFNs on autophagy induction, M-Macs (Donor 111~114) were transfected miRCtrl or miRAB40 in the presence or absence of 1 μg/mL of B18R for three days and then autophagy induction was detected using the Cyto-ID Autophagy detection kit as described in the Materials and Methods. (**F**) Autophagy inducing activities show fold change (FC) compared to that in untreated cells using means of FC ± SE from four independent assays. Each assay was conducted in quadruplicate. ND: not detected, NS: not significant.

**Table 1 ijms-22-01290-t001:** List of novel miRNAs.

miRNA ID	Accession # *	Mature Sequence 5′-3′	Precursor Genomic Location
miRAB1	MF281454	cagagagaggaagagagcugcu	chr3:38142537..38142613:+
miRAB2	MF281455	auguagugcuucuugggacuga	chr3:142310544..142310610:−
miRAB3	MF281456	aauugagguuuuaucugaggggau	chr11:94184662..94184742:+
miRAB4	MF281457	agcacauuuaggaauaggggaa	chr9:5509899..5509964:−
miRAB5	MF281458	ucuauguauggauauguguguau	chr8:22688994..22689070:−
miRAB6	MF281459	agggucucacuguugccagga	chr8:11864247..11864305:−
miRAB7	MF281460	ccaaggucugacucauggguaga	chr7:32869838..32869887:−
miRAB8	MF281461	ucuaccccagggagaaucugaga	chr12:13216143..13216224:+
miRAB9	MF281462	aaggaggaagacugggcauagu	chr5:132487413..132487471:−
miRAB10	MF281463	cugccugugugggacugagaugc	chr6_GL000252v2_alt:2526871..2526961:−
miRAB11	MF281464	aagaguaauugugguuuuuguu	chr4:8218466..8218522:+
MiRAB12	MF281465	uagcccuuccggauccugcgc	chr2:191013985..191014043:−
miRAB13	MF281466	uuuguuuucuaguuaccucu	chr11:6094879..6094943:+
miRAB14	MF281467	ucagggaacagcaacagggcugc	chr1:159800217..159800282:−
miRAB15 **	KY994057	aggacuggaugucgggcugcaugu	chr4:141707940..141708015:+
miRAB16 **	KY994062	acguggacuccagacucucugu	chr17:42311190..42311258:+
miRAB17	MF281468	aaggguuugggucugagcuguau	chr2:85318324..85318388:−
miRAB18	MF281469	cuccccugauguaccugaacaagag	chr3:27374770..27374839:−
miRAB19	MF281470	agcggaacuugaggagccgaga	chr1:161446654..161446715:−
miRAB20	MF281471	auccuagcuugccugagacugu	chr2:71526827..71526913:+
miRAB21	MF281472	ugacuuguguuucuuuuucccaa	chr4:70946713..70946766:−
miRAB22	MF281473	uuagagcuucaaccuccaguguga	chr8:115534466..115534532:−
miRAB23	MF281474	guuaggucaagguguagcccaug	chr5:80651341..80651395:−
miRAB24	MF281475	cuuccccacccucuccugcagc	chr19:12952627..12952689:+
miRAB25	MF281476	cucgggcgcuccggcuguaagg	chr5:96936079..96936137:+
miRAB26	MF281477	agacggaucaggcucuccuc	chr1:44805291..44805352:+
miRAB27	MF281478	ucaguaagagugggcucugucga	chr3:196326926..196326980:−
miRAB28	MF281479	uccggauccagcuucuugucu	chr22:37878013..37878073:+
miRAB29	MF281480	ugcugugugauuuugagugac	chr5:151255955..151256014:−
miRAB30	MF281481	uguccuugggccucuuuguuu	chrX:109740259..109740316:+
miRAB31	MF281482	uuuguacuccggccgcugcuga	chr2:42092534..42092630:+
miRAB32	MF281483	aaagagcuuggucuuuggagcca	chr10:71422254..71422310:+
miRAB33	MF281484	cucggccuuugcucgcagcacucg	chr15:56365282..56365340:−
miRAB34	MF281485	ucuauuucuuguguucccugc	chr8:130528069..130528129:−
miRAB35	MF281486	aucugcucgccggagcucacucu	chr15:75648027..75648082:+
miRAB36	MF281487	cgcgcucgcgggggcucugaga	chr1:178725112..178725166:−
miRAB37	MF281488	cugaggagaacaggagcucucuu	chr22:22085831..22085887:−
miRAB38 **	MF281431	uccccaguacccccacca	chr1:156887613..156887686:+
miRAB39	MF281489	uccauaucccaaccugucagagu	chr10:89197656..89197715:+
miRAB40	MF281490	uaacuucucauuaugccuucugga	chr1:221076658..221076720:+
miRAB41	MF281491	ucaggggaugggagugacauggc	chr17:74522904..74522966:−

*: GenBank Accession number, **: miRNAs previously reported [29,30].

**Table 2 ijms-22-01290-t002:** Expression levels of the predicted genes for each miRNA in the gene expression in Microarray analysis *.

miRNA ID	Gene Name **	Symbol	FC ***	*p* Value ****
miRAB4	dipeptidyl peptidase 4	DPP4	−3.96	*p* < 0.001
miRAB9	RAB6B, member RAS oncogene family	RAB6B	−2.14	*p* < 0.05
	myosin ID	MYO1D	−2.96	*p* < 0.01
miRAB12	actin binding LIM protein family member 3	ABLIM3	−2.29	*p* < 0.01
miRAB14	C-X-C motif chemokine ligand 9	CXCL9	15.08	*p* < 0.05
	RAB6B, member RAS oncogene family	RAB6B	−2.14	*p* < 0.05
miRAB29	dipeptidyl peptidase 4	DPP4	−3.96	*p* < 0.01
	RAB6B, member RAS oncogene family	RAB6B	−2.14	*p* < 0.05
miRAB30	Podocalyxin-like	PODXL	−2.45	*p* < 0.05

*: Microarray analysis was conducted using aliquots of RNA from AB- and ABI-Mac of Donor 89~91 that was used in miRSeq (Figure 1). **: Predicted genes for each miRAB (Figure 3) was cross-analyzed with microarray data. Common genes significantly regulated by fold change (FC) > 2, *p* values < 0.05) are listed. ***: FC in the gene expression in ABI-Mac compared to that in AB-Mac in microarray analysis. ****: *p* values were obtained from 2-way ANOVA.

**Table 3 ijms-22-01290-t003:** Expression levels of HDF and ARF in miRAB40-transfeced cells.

**Host-Dependency Factors (HDFs) for HIV Infection**		
**Gene**	**FC ***	***p* Values ****
HIBCH	1.7 ± 0.2 (4)	*p* < 0.01
TBL1X	1.5 ± 0.2 (4)	*p* < 0.05
SLC24A1	2.5 ± 0.5 (4)	*p* < 0.05
MAP3K7	1.8 ± 0.4 (4)	NS ***
ETS2	2.0 ± 1.2 (4)	NS
DHX15	1.5 ± 0.2 (4)	*p* < 0.05
**Autophagy Regulatory Factors (ARFs)**		
**Gene**	**FC**	**p Values**
PIK3C	1.6 ± 0.2 (4)	*p* < 0.05
RAB9A	2.5 ± 0.6 (4)	n.s
NRAS	1.3 ± 0.2 (4)	n.s
SIK1	5.2 ± 1.7 (4)	n.s
ITGb1	1.1 ± 0.1 (4)	n.s
CIP2A/KIAA1524	1.0 ± 0.2 (3)	n.s

* Fold change in increase in the gene expression in miRAB40-transfected M-Mac (Donor 119~121) compared to that in miRCtrl-transfected M-Mac by real time RT-PCR. Results show mean ± SE (*n* = 3). **: *p* values were calculated using unpaired Student *t*-test, *** not significant.

## Data Availability

The sequencing Data have been deposited to SRA database at National Center for Biotechnology Information (https://www.ncbi.nlm.nih.gov/bioproject/PRJNA649982/).

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
