# Peer review of "MicroRNA Profiles in Monocyte-Derived Macrophages Generated by Interleukin-27 and Human Serum: Identification of a Novel HIV-Inhibiting and Autophagy-Inducing MicroRNA"

_ijms, 2021, doi:10.3390/ijms22031290_

Round 1

Reviewer 1 Report

Manuscript: ijms-1055542

Title: MicroRNA Profiles in Monocyte-derived Macrophages Generated by
Interleukin-27 and Human Serum: Identification of a Novel HIV-inhibiting and
Autophagy-inducing MicroRNA

Overall:

In this manuscript by Imamichi T et al, the effect of IL-27 on macrophage differentiation and expression of microRNAs in the context of HIV replication was studied.  The authors identified 38 novel miRNAs with miRAB40 inhibiting mRNA transcription of HIV in a Type I IFN dependent authophagicitic manner.  The authors data are significant and novel and help move the miRNA and HIV fields forward.  In order verify and clarify the authors conclusions, the human donors and their characteristics need to be presented, as well as, which donor’s were used in each experimental design.  The authors also need to clarify the results between the AB-, ABI- and M-Macs as the miRNAs identified in the AB- and ABI-Macs need to be verified in the M-Macs (initial screen) as well as I-Macs, and HIV donor macs.      

Major Comments:

  • The authors perform experiments on human donor cells but change between a variety of in vitro stimulation conditions: AB-, ABI-Mac, M-Mac, plus/minus HIV virus (HIVAD8 or IVLuc-V), or transfected miRNA (miRAB40 or miRCtrl). Its unclear which donors (HIV+ vs HIV-) were used during which experiments making comparisons difficult and complicated.  Were HIV+ donors MDMs or I-Macs used to verify the in vitro generated miRNA data ? Also, the switch between AB- and M-Macs provides a proof-of-concept; however, the results should be verified in both cell lines and in I-Macs which might be similar to ABI-Macs. For example, were ABI-Macs A) resistant to HIV viral infection (HIVAD8 or IVLuc-V), B) increased in autophagy, and c) increased in IFN-α and -β.
  • A significant number of miRNAs as show in Figure 4A, enhanced HIV replication. An analysis or discussion of these would expand the authors conclusion and possibly shed light on novel mechanisms of miRNA discussed in the introduction of the manuscript (lines 44-45).  
  • miRAB3 does not appear to mechanistically use autophagy and would A) provide a nice control for the IFN, and HDFs and ARFs experiments as well as help elucidate novel mechanisms of HIV regulation.

Specific Comments:

  • To help increase the significance, the abstract should focus on the miRNA detection and similarities and decrease the mechanistic aspect of miRAB40 as it remains unknown (remove or change lines 24-28).
  • Figure 1 legend. There are formatting errors in the legend (lines 102-105).
  • The authors state that 3 of their previously identified 3 miRNAs were not updated into miRbase 22.1 (lines 116-118). Therefore, is it also possible that the remaining 38 miRNAs identified in this study could have been identified elsewhere and not updated in miRBase. Are there additional micro-RNA databases to mine ?
  • Figure 2. What statistical test was performed to generate the listed p values ? A vin diagram like Figure 1 for these 5 new donors might provide useful data. A proof of concept analysis performed for these 5 new donors like Figure 5 for all miRNA expression would provide a way to observe the overlap in these miRNAs in a pool of 8 donors.
  • Figure 3/Table 2. Please clarify the donor population used for the target prediction analyses. N=3 is presented but in total there are N=8 donors ? If all 8 donors were used, does the genes that are significant based on fold-change replicate or differ ?
  • Figure 4. While the authors looked at inhibition of HIV replication by the novel miRNAs (#3 and #40), increases in replication were also observed (#17, 20, 41, 2) and appear to be more predominant then those that inhibited. The authors should discuss this. Please clarify if these donors were the same donors from the initial miRNA studies. Please define the protein/label used to detect autophagy in the IF images within the legend.
  • Figure 5. Please define the donors used in these assays. Were additional miRNAs that did not inhibit HIV tested in these assays as they would provide a useful control when interpreting the data.
  • Table 3. The authors propose that Host-dependency factors (HDFs) and/or Autophagy regulator factors (ARFs) are targets for the novel miRNAs identified in this study. The authors transfect miRAB40, a single identified miRNA, into AB- or ABI-Mac and look for mRNA expression. However, the authors did not detect any significant changes (minus an increase in SLC24A1) and no protein changes.  Was the initial AB- and ABI-Mac treated donors screened for all the HDFs and ARFs to observe mRNA changes or the M-Mac cells used to perform the initial HIV infected screens ? In addition were miRAB40 transfected M-Mac cells used for HDF/ARF detection ?
  • Materials and Methods – Please describe the number of donors used in each experiment. A demographics Table for the donors would also help illustrate the heterogeneity of the human population. In the Generation of Cells (line 387) please define all conditions used including M-MAC, HIV infected, miRNA transfected and define which were used to isolate miRNA and perform individual types of experiments.
  • Autophagy images were captured at 10x (line 501); however, a higher power magnification might produce a higher quality image to analyze. Also, 50x103 cells line 507 should be expressed as 50x103
  • Statistical analysis – is a Student’s t test appropriate for all the data sets ? An ANOVA is more appropriate for data sets of 3 or more, as an example.
  • Unfortunately, I did not have access to the supplemental materials and therefore could not comment.

Author Response

Point-by-point Response letter

Reviewer #1

Overall:

In this manuscript by Imamichi T et al, the effect of IL-27 on macrophage differentiation and expression of microRNAs in the context of HIV replication was studied.  The authors identified 38 novel miRNAs with miRAB40 inhibiting mRNA transcription of HIV in a Type I IFN dependent authophagicitic manner.  The authors data are significant and novel and help move the miRNA and HIV fields forward.  In order verify and clarify the authors conclusions, the human donors and their characteristics need to be presented, as well as, which donor’s were used in each experimental design.  The authors also need to clarify the results between the AB-, ABI- and M-Macs as the miRNAs identified in the AB- and ABI-Macs need to be verified in the M-Macs (initial screen) as well as I-Macs, and HIV donor macs.   

Major Comments:

  1. The authors perform experiments on human donor cells but change between a variety of in vitro stimulation conditions: AB-, ABI-Mac, M-Mac, plus/minus HIV virus (HIVAD8 or IVLuc-V), or transfected miRNA (miRAB40 or miRCtrl). Its unclear which donors (HIV+ vs HIV-) were used during which experiments making comparisons difficult and complicated.  Were HIV+ donors MDMs or I-Macs used to verify the in vitro generated miRNA data ? Also, the switch between AB- and M-Macs provides a proof-of-concept; however, the results should be verified in both cell lines and in I-Macs which might be similar to ABI-Macs. For example, were ABI-Macs A) resistant to HIV viral infection (HIVAD8 or IVLuc-V), B) increased in autophagy, and c) increased in IFN-α and -β.

Response: (Please read our response to comment 6 of Reviewer #2). In order to clarify the donor usage in each assay, we have now inserted a diagram in each figure (Figure1A, 2A, 4A, 4C, 5A, 5C, and 5E and each figure legend has been inserted in lines 103~105, 146~147, 215~216, 221~223, 261~262, 264~265,and 269~272) and provided Donor numbers in the text lines 181, 191, 250, 324, 329, 338, and 379. Since the diagrams were inserted, original figure numbers have been changed as followed:

     Original MS       Revised MS     

     Figure 1A~C     Figure 1B~D

     Figure 2            Figure 2B

     Figure 4A          Figure 4B

     Figure 4B          Figure 4D

     Figure 4C         Figure 4E

     Figure 5A          Figure 5B

     Figure 5B          Figure 5D

     Figure 5C         Figure 5F

As we described in the original manuscript (MS) in the Materials and Methods line 388~389, all experiments were performed cells from healthy donors  who were not infected with HIV. The statement has now clearly incorporated in lines 86 and 438~439.

The current study was aimed to define the profiles of novel microRNAs in ABI-Mac and to identify/characterize biological functions of the novel miRNAs. It would be an interest to analyze microRNA profiles in HIV-infected patients (along with plasma viral load); however, the study is beyond our current goal. We greatly appreciate the reviewer’s comment. We would consider incorporating the study in our future study. In response to the reviewer’s comment, we have incorporated the statement in lines 414~417.

As described in the introduction of the original manuscript (MS) line 65-69, we have previously investigated miRNA profiles in M-Mac and I-Mac, however, the 38 novel miRNAs discovered in the current study were not detected in our previous study, thus it was plausible that the novel miRNAs may not express in M-Mac. In response to the comment, we have incorporated a statement in the discussion section lines 349~354.

As we described in the introduction of the original MS lines 75-76 (lines 76-77 in the revised MS), we have reported that ABI-Mac resists to infection of HIV AD8 and HIVLuc-V and induces autophagy [ref 28] (a MS describing the mechanism of HIV-resistance and autophagy induction is currently under revision in a journal) and based on our previous observation, in order to further characterize ABI-Mac, we conducted the current study focusing on identification of miRNA profiles.

  1. A significant number of miRNAs as show in Figure 4A, enhanced HIV replication. An analysis or discussion of these would expand the authors conclusion and possibly shed light on novel mechanisms of miRNA discussed in the introduction of the manuscript (lines 44-45).  

Response: We agree with the reviewer that some miRNAs enhanced HIV replication or infection by less than 1.5-fold (except miRAB17, miRAB20 and miRAB41 which enhanced HIVAD8 replication by near 2-fold), however, a statistical analysis resulted in that the effect  was not significant, thus we did not focus on those miRAB in the current study. In response to the comment, we have input a statement in the result section lines 209-210, and discussed it in lines 382-387 in the revised manuscript

  1. miRAB3 does not appear to mechanistically use autophagy and would A) provide a nice control for the IFN, and HDFs and ARFs experiments as well as help elucidate novel mechanisms of HIV regulation.

Response: We agree with the reviewer that miRAB3 may be one of a good control for elucidating the mechanism and we described it in the discussion section of the original MS line 337-341 (lines 378~381 in the revised MS); however, as we described above (our response to comment 1), the goal of the current study was to identify and characterize novel miRNAs, thus we did not purse the mechanism in the current study. miRAB3 and other would be used as controls in our future studies. 

Specific Comments:

  1. To help increase the significance, the abstract should focus on the miRNA detection and similarities and decrease the mechanistic aspect of miRAB40 as it remains unknown (remove or change lines 24-28).

Response: We modified the abstract accordingly in lines 24-25.

  1. Figure 1 legend. There are formatting errors in the legend (lines 102-105).

Response: Thank you for the comment. We have corrected it in lines 104-107.

  1. The authors state that 3 of their previously identified 3 miRNAs were not updated into miRbase 22.1 (lines 116-118). Therefore, is it also possible that the remaining 38 miRNAs identified in this study could have been identified elsewhere and not updated in miRBase. Are there additional micro-RNA databases to mine ?

Response:  We had researched the GenBank and the miRbase before submitting the MS and we did not see any identical miRNAs with our 38 novel miRNAs in both data banks.  As the reviewer pointed out, we blasted our novel miRNA in response to the comment. As of 12/31/2020, none of our novel miRNAs are identical with reported  miRNA (to the best our knowledge, the miRbase is the only official miRNA databank).  We have now incorporated the statement in lines 121-123.

  1. Figure 2. What statistical test was performed to generate the listed p values ? A vin diagram like Figure 1 for these 5 new donors might provide useful data. A proof of concept analysis performed for these 5 new donors like Figure 5 for all miRNA expression would provide a way to observe the overlap in these miRNAs in a pool of 8 donors.

Response: In Figure 2, we have used the unpaired Student t-test for results from qRT-PCR to confirm the expression of novel miRABs as described in our original Materials and Methods section, line 519~523. We have modified the Materials and Methods section in lines 579~584 (please see our response to comment 3 from the Reviewer 2). The goal of Figure 2 is to elucidate whether the novel miRNAs express in general without a donor dependency, thus we used RNA from five new donors (donor 92~96).  Figure 1 indicates a Venn diagram focusing on known miRNAs from miRSeq data. As the reviewer commented, a proof of concept analysis may provide a way to observe the overlap, however it is beyond our goal of this project (the identification of novel miRNAs). We greatly appreciate the comment, and we will consider the analysis in future project. To clearly describe the goal of Figure2, we have modified text in lines 135~138.

  1. Figure 3/Table 2. Please clarify the donor population used for the target prediction analyses. N=3 is presented but in total there are N=8 donors ? If all 8 donors were used, does the genes that are significant based on fold-change replicate or differ ?

Response:  Figure 3 shows results from prediction analyses  for each miRAB using four prediction tools as mentioned in the original Materials and Methods line 432~445 (the revised MS in line 486~499). The tools predict potential target genes for each miRNA using human gene data base, not our donor samples. To clearly describe the purpose of the analysis, the text was modified (lines 164~165).

Table 2 shows a result from a cross analysis between Microarray analysis and  miRSeq using the same RNA from  thee three donor mRNA as described in the original text line 447.  To clearly describe the table, we have modified the title of the Table (line 190) and the footnotes to clearly describe the analysis in lines 191-194.

  1. Figure 4. While the authors looked at inhibition of HIV replication by the novel miRNAs (#3 and #40), increases in replication were also observed (#17, 20, 41, 2) and appear to be more predominant then those that inhibited. The authors should discuss this.

Response: Please see our response to the comment 2 above.                                           

Please clarify if these donors were the same donors from the initial miRNA studies.

Response: The donors were different from the initial mRNA studies. To clearly describe the donor differences, we have inserted a diagram in each figure (a new Figure 4A and the original figure 4A is now 4B) and incorporated the statement in the figure legend.  

Please define the protein/label used to detect autophagy in the IF images within the legend.

Response: We have modified the legend accordingly (lines 225-228).

  1. Figure 5. Please define the donors used in these assays. Were additional miRNAs that did not inhibit HIV tested in these assays as they would provide a useful control when interpreting the data.

Response:  In these assays, the same lot of M-Mac from the donors for Figure 5B was  used in Figure 5F.  All donor usages were incorporated in the figure legend in line 261, 264~265, and 272. The miRCtrl has been used as a control in the assay.

  1. Table 3. The authors propose that Host-dependency factors (HDFs) and/or Autophagy regulator factors (ARFs) are targets for the novel miRNAs identified in this study. The authors transfect miRAB40, a single identified miRNA, into AB- or ABI-Mac and look for mRNA expression. However, the authors did not detect any significant changes (minus an increase in SLC24A1) and no protein changes. 

Response:  We have not transfected miRAB40 in AB or ABI-Mac in this study. As described in the original MS at  lines 188~189 (in the revised MS, the line 199 ~200), all biological studies have been done using M-Mac.  To clearly describe the context in Table 3, we have modified the footnote (lines 338~339).

Was the initial AB- and ABI-Mac treated donors screened for all the HDFs and ARFs to observe mRNA changes or the M-Mac cells used to perform the initial HIV infected screens ?

Response:  We have screened HDF and ARF in ABI-Mac and AB-Mac in another study (currently a MS describing the results is under revision), we did not observe significant changes in ARFs [28]. The goal of the current study was to identify novel miRNA in those cells, and the profiles studies of HDF or ARF is beyond the goal, therefore we have not described them.

In addition were miRAB40 transfected M-Mac cells used for HDF/ARF detection ?

 Response:  The table 3 shows the results from a comparison study of the expression of HDF and ARF that potentially targeted by miRAB40 in the miRCtrl- or the miRAB40-transfected M-Mac. To clearly demonstrated the result,  we have modified the text in lines 323~324 and 329~330 and revised the footnote of the table in line 338~339.

  1. Materials and Methods – Please describe the number of donors used in each experiment. A demographics Table for the donors would also help illustrate the heterogeneity of the human population. In the Generation of Cells (line 387) please define all conditions used including M-MAC, HIV infected, miRNA transfected and define which were used to isolate miRNA and perform individual types of experiments.

Response:  (Please see our response to reviewer 2, comment 6).  We have inserted a diagram in each figure in the revised MS and described donor IDs,  instead of providing a demographics Table. As can been seen in Figure 5, to directly define a correlation between IFN induction and autophagy induction, we used aliquots of same M-Mac from three different donors (Donor 111~114) in the assays.

  1. Autophagy images were captured at 10x (line 501); however, a higher power magnification might produce a higher quality image to analyze. Also, 50x103 cells line 507 should be expressed as 50x103

Response: As described in the original Materials and Methods (lines 493-507), we have used the FiJi Image J software to quantify autophagy induction in individual cell. We have now clearly described the assay in results section lines 233~236 and in the method section line 559. Autophagy images in the new Figure 4E (original Figure 4C) are provided to visually demonstrate readers the significant difference in the autophagy induction. We have incorporated the status in lines 240~241.

The method section was corrected by deleting the sentence.

  1. Statistical analysis – is a Student’s t test appropriate for all the data sets ? An ANOVA is more appropriate for data sets of 3 or more, as an example.

Response: In biological functional study, we compared activity of HIV inhibition, autophagy induction or IFN induction between two parameters (miRCtrl- and miRAB mimic-transfected M-Mac) and each assay was replicated multiple times; therefore, Student’s t-test is appropriate method. ANOVA is an appropriate method for three or more parameters, and we have used it for Microarray analysis as described in the original MS line 451~453 (lines 507-509 in the revised MS).

  1. Unfortunately, I did not have access to the supplemental materials and therefore could not comment.

Response: When we submitted our original MS with supplemental materials, we had not been noticed any misloading; our original submission was passed a quality test. We are sorry for inconvenient for accessing the materials under review.

Reviewer 2 Report

The manuscript is very valuable. However, some points should be considered once again and included.

  1. Please provide the bioethics commission approval number and date of obtaining it.
  2. Abbreviations should be explained when they are first used.
  3. The statistical analysis (4.16.) has to be described in more details.
  4. Explaining what normality test was used to confirm the use of parametric methods.
  5. Please add the reason of using GAPDH as an endogenous control, not either ACTB or 18S rRNA.
  6. Please provide a diagram (figure) of the individual stages of the experiment.
  7. References and work layout should be adapted to the requirements of the journal.
  8. Please try to replace the active voice (we) with the passive voice.
  9. Please show the value of the miRSVR parameter for miRNAs regulating expression of the IL-27.

Author Response

Point-by-point Response letter

Reviewer #2

Comments and Suggestions for Authors

The manuscript is very valuable. However, some points should be considered once again and included.

  1. Please provide the bioethics commission approval number and date of obtaining it.

Response: The number has been incorporated in the Materials and Methods in lines 435~436.

  1. Abbreviations should be explained when they are first used.

Response: We have changed all abbreviations in the text accordingly (lines 58,121,176, 178, 188, 202, 252, 313~316, 319~322, 343~344, 389~390, 443, 495, 505, 526, and 539~540).

  1. The statistical analysis (4.16.) has to be described in more details.

Response: We have provided more detailed method in line 579~584.

  1. Explaining what normality test was used to confirm the use of parametric methods.

Response: In miRNA seq analysis, we have used EdgeR, which is based on the negative binomial distribution was used to prior to the data analysis.  In microarray analysis, we used RMA normalization for the data, prior to the data analysis. We have incorporated the statement in the Materials and Methods section in lines 471~474 and 504~506.

  1. Please add the reason of using GAPDH as an endogenous control, not either ACTB or 18S rRNA.

Response: We are aware of  that some publications mention that GAPDH may not be suitable internal control because the expression of the gene may change among samples; however, our group routinely uses GAPDH as the internal control for qRT-PCR  in vitro assay and used it in this study to obtain comparable data with previous works.  By our hands, Ct values of GAPDH consistently indicate around 20~21 without any drastic changes in all assays. Therefore, we have been used the gene as an internal control. In response to the reviewer’s comment, we have incorporated a statement in the Materials and Methods in line 527~528.

  1. Please provide a diagram (figure) of the individual stages of the experiment.

Response: Diagrams for the individual stages of the experiment have been incorporated Figure1A, 2A, 4A, 4C, 5A, 5C, and 5E,  and each figure legend has been inserted in lines 103~105, 146~147, 215~216, 221~223, 261~262, 264~265,and 269~272. Since the diagrams were inserted, original figure numbers have been changed as followed:

     Original MS       Revised MS     

     Figure 1A~C     Figure 1B~D

     Figure 2            Figure 2B

     Figure 4A          Figure 4B

     Figure 4B          Figure 4D

     Figure 4C         Figure 4E

     Figure 5A          Figure 5B

     Figure 5B          Figure 5D

     Figure 5C         Figure 5F

  1. References and work layout should be adapted to the requirements of the journal.

Response: The layout of MS was revised accordingly.

  1. Please try to replace the active voice (we) with the passive voice.

Response: The entire manuscript was revised using the passive voice accordingly (lines 17~18, 73~74, 75~76, 95,109~110, 116~117, 123~124, 196, 277~278, 358~359, 418, and 420).

  1. Please show the value of the miRSVR parameter for miRNAs regulating expression of the IL-27.

Response: This study was aimed to compare the miRNA expression profiles between AB-Mac and ABI-Mac and to define how IL-27 regulates miRNA expression in the cells; however, we have not investigated function of miRNA that regulates IL-27 expression as the reviewer’s comment, therefore, we were not able to conduct the requested analysis at this moment. We have modified abstract and introduction to clearly describe the goal of this study in lines 17~18 and 80~82.   

Round 2

Reviewer 1 Report

Manuscript: ijms-1055542-r1

Title: MicroRNA Profiles in Monocyte-derived Macrophages Generated by
Interleukin-27 and Human Serum: Identification of a Novel HIV-inhibiting and
Autophagy-inducing MicroRNA

Overall:

In this revised manuscript by Imamichi T et al, the effect of IL-27 on macrophage differentiation and expression of microRNAs in the context of HIV replication was studied.  The authors identified 38 novel miRNAs with miRAB40 inhibiting mRNA transcription of HIV in a Type I IFN dependent authophagicitic manner.  The authors data are significant and novel and help move the miRNA and HIV fields forward.  While the authors have improved the manuscript in general, there remain significant concerns in the reproducibility of the authors conclusions to a larger population.  Beginning with Figure 4, the authors continually switch between ‘independent’ donors and conclude the results will be similar amongst the populations tested. For example, in both Figure 4 and 5 the donors used for HIV infection (Fig 4B) is not the same donor pool as the data for Autophagy (Fig 4D). This issue is similar in Figure 5 where the donor for the HIV infection and IFN response is different than the autophagy and IFN response which are both different from the donor population in Figure 4 and the original population used to identify the miRNA (Figure 1 and 2).  In this case, a demographics Table describing the donor populations in terms of age, sex, gender and race are critical to at least account for heterogeneity and variability in donor to donor differences.  The authors also try to connect the dots between identification of novel miRNAs in AB- and ABI-Macs to inhibition of HIV replication in a possibly IFN and Host-dependency factors (HDFs) and/or Autophagy regulator factors (ARFs) mechanism.  These conclusions are a stretch of the data presented with significant concerns that donor to donor variability will impact the interpretation of the results.  

Major Comments:

  • The authors perform experiments on human donor cells but change the donor population between experimental conditions and assays. Were the same experimental conditions performed across the entire donor population, ie Donors 89-121 ?  As experimental conditions were performed with an independent set of donors (N=3 to 5) each time, the authors conclusions are limited to those specific donors and assays and may not apply across a generalized population. 
  • As independent donors were used in each experiment, the authors conclusions could in part be explained by donor to donor heterogeneity. In order to increase the confidence in the data, the same donor pool should be analyzed across all the experiments.

Specific Comments:

  • Figure 2. In A) Should M-Mac be changed to AB-Mac or ABI-Mac ? An important question is how similar in expression are the unique miRNAs discovered in Figure 1 across the entire donor population used throughout the manuscript (ie., Donors 89-121). Presenting a table with the level of expression of the 38 unique miRNAs across the Donors 89-121 would be useful.  
  • Figure 5. In addition to the control miRNAs, miRNAs that did not inhibit HIV tested (Fig 4) would increase the significance of these findings.
  • Table 3. The authors propose that Host-dependency factors (HDFs) and/or Autophagy regulator factors (ARFs) are targets for the novel miRNAs identified in this study. The authors transfect miRAB40, a single identified miRNA, into M-Mac and look for mRNA expression. However, the authors did not detect any significant changes (minus an increase in SLC24A1) and no protein changes.  This is representative of a single miRNA, as miRAB3 was not tested, in 3 representative donors after M-Mac generation. The effect in these donors to miRAB40 to HIV infection and autophagy (Fig 4 and Fig 5) should be presented as there is likely donor to donor variability in these types of assays.
  • The authors reply to the dataset in Table 3 was as follows:  
    1. The authors reply in the revision: “We have screened HDF and ARF in ABI-Mac and AB-Mac in another study (currently a MS describing the results is under revision), we did not observe significant changes in ARFs [28]. The goal of the current study was to identify novel miRNA in those cells, and the profiles studies of HDF or ARF is beyond the goal, therefore we have not described them.” In the abstract “We recently reported that macrophages differentiated from monocytes in the presence of IL-27 and human AB serum resisted human immunodeficiency virus (HIV) infection and showed significantly enhanced autophagy. In the current study, we investigated the miRNA profiles of these cells., especially focusing on novel miRNAs.” The authors go on to analyze and detail in the manuscript the function of these novel miRNAs on HIV replication and propose a mechanism via Autophagy and possibly IFN signaling. 
    2. The authors try to connect the dots between AB serum and IL-27 on HIV resistance via novel miRNAs. If there is AB- and ABI-Mac data suggesting these HDF and ARF targets are present and/or altered in AB- and ABI-Mac treated donors it would significantly strengthen the manuscript. If this data is not available or the authors do not wish to include it, then Table 3 should be moved to the supplemental figures and presented only as part of the discussion in the manuscript as the authors state above this was not the goal of the manuscript.
  • Autophagy images were captured at 10x (Fig 4); however, a higher power magnification would help illustrate the structural and visual components of Autophagy in these cells. While the images were quantitated at 10X and there is a notable difference in positive vs negative, its hard to define any cellular features in the image.

Author Response

Point-by-point response

Major Comments:

  • The authors perform experiments on human donor cells but change the donor population between experimental conditions and assays. Were the same experimental conditions performed across the entire donor population, ie Donors 89-121 ?  As experimental conditions were performed with an independent set of donors (N=3 to 5) each time, the authors conclusions are limited to those specific donors and assays and may not apply across a generalized population. 

Response: Yes, all experiments were performed under the same condition. In this study, we report the detection of novel miRNAs in the setting of our samples. We do understand that If we report  the expression-level in general, we need to perform an analysis using robust sample size;  however, that is not our goal in this study. We tried to report that  the detection of  the novel miRNAs in the setting of our sample size and the characterization of potential dual functions in miRAB40. The expression levels of each miRNA were not precisely investigated it in this study, thus we have not reported in the MS.  To clearly describe that we found a potential bifunctional miRNA in the cells, we have modified Abstract and the text in lines 19, 26, 96~101, 109, 113~115, 126, 148~150, 255, 362~364,367~376, 378, 436~439, 445~448.  

  • As independent donors were used in each experiment, the authors conclusions could in part be explained by donor to donor heterogeneity. In order to increase the confidence in the data, the same donor pool should be analyzed across all the experiments.

Response: In this study, we try to report 1) a discovery of novel miRNA that were detected in AB and IL-27-induced macrophages (we did not focus on expression level of each miRNAs in individual)  and 2) miRAB40 is HIV inhibiting and autophagy inducing miRNA. We understand the reviewer’s point; in our previous version of MS, we tried to report our finding as a general one. understand that our previous version of MS describes as if  our findings were general ones, but we shouldn’t.  We have modified the context in the text lines as described above.

To clearly describe our goal, we replaced  a word “expression” to “detection”. To avoid misreading our finding as a general finding,  now we incorporated “potential” in the text.

Specific Comments:

  • Figure 2. In A) Should M-Mac be changed to AB-Mac or ABI-Mac ? An important question is how similar in expression are the unique miRNAs discovered in Figure 1 across the entire donor population used throughout the manuscript (ie., Donors 89-121). Presenting a table with the level of expression of the 38 unique miRNAs across the Donors 89-121 would be useful.  

Response:  Thank you for the comment, Figure A has been revised. Please see our response above. We did not try to compare the expression level in each miRNA in this study, we report the detection of the genes without considering the expression level. 

  • Figure 5. In addition to the control miRNAs, miRNAs that did not inhibit HIV tested (Fig 4) would increase the significance of these findings.

Response:  In this study, we report a potential function of miRAB40, we will consider more detail analysis to increase the significancy. In response to the comment, we have incorporated the statement in lines 395~397.

  • Table 3. The authors propose that Host-dependency factors (HDFs) and/or Autophagy regulator factors (ARFs) are targets for the novel miRNAs identified in this study. The authors transfect miRAB40, a single identified miRNA, into M-Mac and look for mRNA expression. However, the authors did not detect any significant changes (minus an increase in SLC24A1) and no protein changes.  This is representative of a single miRNA, as miRAB3 was not tested, in 3 representative donors after M-Mac generation. The effect in these donors to miRAB40 to HIV infection and autophagy (Fig 4 and Fig 5) should be presented as there is likely donor to donor variability in these types of assays.

Response: Please read our response above.  We modified the text. We modified our statement that we found that miRAB40 is a potential anti-HIV and autophagy inducing miRNA.

  • The authors reply to the dataset in Table 3 was as follows:  
    1. The authors reply in the revision: “We have screened HDF and ARF in ABI-Mac and AB-Mac in another study (currently a MS describing the results is under revision), we did not observe significant changes in ARFs [28]. The goal of the current study was to identify novel miRNA in those cells, and the profiles studies of HDF or ARF is beyond the goal, therefore we have not described them.” In the abstract “We recently reported that macrophages differentiated from monocytes in the presence of IL-27 and human AB serum resisted human immunodeficiency virus (HIV) infection and showed significantly enhanced autophagy. In the current study, we investigated the miRNA profiles of these cells., especially focusing on novel miRNAs.” The authors go on to analyze and detail in the manuscript the function of these novel miRNAs on HIV replication and propose a mechanism via Autophagy and possibly IFN signaling. 
    2. The authors try to connect the dots between AB serum and IL-27 on HIV resistance via novel miRNAs. If there is AB- and ABI-Mac data suggesting these HDF and ARF targets are present and/or altered in AB- and ABI-Mac treated donors it would significantly strengthen the manuscript. If this data is not available or the authors do not wish to include it, then Table 3 should be moved to the supplemental figures and presented only as part of the discussion in the manuscript as the authors state above this was not the goal of the manuscript.

Response: In this study, we used ABI-Mac to detect novel miRNAs, but we did not try to make a connection between the HIV-resistance in ABI-Mac and the expression of  novel miRNA in ABI-Mac (Table 3 is a result in miRAB40-transfected cells). We are currently preparing a MS to report the correlation among miRNAs and anti-HIV and autophagy induction using multiple cell types. To demonstrate the  expression of HDF and ARF in AB- and ABI-Mac is beyond current goal.

  • Autophagy images were captured at 10x (Fig 4); however, a higher power magnification would help illustrate the structural and visual components of Autophagy in these cells. While the images were quantitated at 10X and there is a notable difference in positive vs negative, its hard to define any cellular features in the image.

Response: Figure 4E has been replaced with zoomed in images accordingly.

Reviewer 2 Report

The authors answered for my comments.

Thank you !

Author Response

Thank you so much.

Round 3

Reviewer 1 Report

Manuscript: ijms-1055542-r2

Title: MicroRNA Profiles in Monocyte-derived Macrophages Generated by
Interleukin-27 and Human Serum: Identification of a Novel HIV-inhibiting and
Autophagy-inducing MicroRNA

Overall:

In this additional revised manuscript by Imamichi T et al, the effect of IL-27 on macrophage differentiation and expression of microRNAs in the context of HIV replication was studied.  The authors identified 38 novel miRNAs with miRAB40 inhibiting mRNA transcription of HIV in a Type I IFN dependent authophagicitic manner.  The authors data are significant and novel and help move the miRNA and HIV fields forward.  The authors have attempted to explain their results are observational and limited to the donors used in each experiment; however, there remain significant concerns in the reproducibility of the authors conclusions to a larger population.  In this case, a demographics Table describing the donor populations in terms of age, sex, gender and race are critical to at least account for heterogeneity and variability in donor to donor differences.  These conclusions are observational based on the donor pool chosen for each set of experiments where donor to donor variability across experiments will impact the interpretation of the results.  

Major Comments:

  • The authors perform experiments on human donor cells but change the donor population between experimental conditions and assays. A demographics Table describing the donor populations in terms of age, sex, gender and race are critical to at least account for heterogeneity and variability in donor to donor differences across the entire donor population, ie Donors 89-121.  As experimental conditions were performed with an independent set of donors (N=3 to 5) each time, the authors conclusions are limited to those specific donors and assays and may not apply across a generalized population.

Specific Comments:

  • Figure 4. Please update the lens used to capture the zoomed in images as its likely no longer 10x.

Author Response

Thank you for your comment.

The current goal of this study is to identify novel miRNAs in the human AB serum induced macrophage with or without IL-27. To define the donor to donor heterogeneity is beyond the current study. We appreciate the comment and would consider in future study.  

The provided images are zoomed in images of 10x, thus there are still 10x.